# Bacterial Surface Disturbances Affecting Cell Function during Exposure to Three-Compound Nanocomposites Based on Graphene Materials

**DOI:** 10.3390/nano12173058

**Published:** 2022-09-02

**Authors:** Agata Lange, Ewa Sawosz, Karolina Daniluk, Mateusz Wierzbicki, Artur Małolepszy, Marcin Gołębiewski, Sławomir Jaworski

**Affiliations:** 1Department of Nanobiotechnology, Institute of Biology, Warsaw University of Life Sciences, 02-786 Warsaw, Poland; 2Faculty of Chemical and Process Engineering, Warsaw University of Technology, 00-654 Warsaw, Poland; 3Department of Animal Breeding, Institute of Animal Sciences, Warsaw University of Life Sciences, 02-786 Warsaw, Poland

**Keywords:** nanocomposites, bacteria, graphene materials, cell functions

## Abstract

Combating pathogenic microorganisms in an era of ever-increasing drug resistance is crucial. The aim of the study was to evaluate the antibacterial mechanism of three-compound nanocomposites that were based on graphene materials. To determine the nanomaterials’ physicochemical properties, an analysis of the mean hydrodynamic diameter and zeta potential, transmission electron microscope (TEM) visualization and an FT-IR analysis were performed. The nanocomposites’ activity toward bacteria species was defined by viability, colony forming units, conductivity and surface charge, cell wall integrity, ATP concentration, and intracellular pH. To ensure the safe usage of nanocomposites, the presence of cytokines was also analyzed. Both the graphene and graphene oxide (GO) nanocomposites exhibited a high antibacterial effect toward all bacteria species (*Enterobacter cloacae*, *Listeria monocytogenes*, *Salmonella enterica*, and *Staphylococcus aureus*), as well as exceeded values obtained from exposure to single nanoparticles. Nanocomposites caused the biggest membrane damage, along with ATP depletion. Nanocomposites that were based on GO resulted in lower toxicity to the cell line. In view of the many aspects that must be considered when investigating such complex structures as are three-component nanocomposites, studies of their mechanism of action are crucial to their potential antibacterial use.

## 1. Introduction

The biochemical reactions of bacterial metabolism constitute the base of their survival. They provide structural and functional components as well as the energy required for life processes [1]. Changes in energy metabolism are, in turn, associated with other cellular processes such as nutrient uptake or the efflux of toxic compounds [2]. Energy dependencies are explained by the commonly known Mitchell’s theory, under which the “intact” membrane serves as a proton pump, in which, during electron transport, a proton extrusion reaction occurs, and energy is generated. Such energy is strictly linked to ATP synthesis [3].

The bacteria’s external surface is crucial in the interaction between the cell and the surrounding environment [4]. Peptidoglycan is one of the basic components of the cell wall structure. Differentiation between Gram-positive and Gram-negative bacteria relies on the various natures of peptides, while the sugar moieties’ compositions are common for both types [5].

Nanomaterials are used as antimicrobial agents because they perform multiple mechanisms that occur simultaneously in bacteria cells, in contrast to traditional antibiotics whose mechanisms are acting as simple structures or processes [6,7,8]. The toxicity of nanomaterials is affected by many different aspects, including size, surface area and chemistry, and the ability to agglomerate [9]. An increase in the proportion of atoms and molecules on a nanomaterial’s surface causes the surface properties to determine their behavior [10]. One possible usage of nanoparticles is their application in the form of drug carriers, given that they have a small size and that they may be potentially biocompatibility if their surface electron status, which is involved in cytotoxicity, is properly prepared [11]. Despite the same scale, distinct types of nanomaterials exhibit different physical and chemical properties. Metal and metal oxide nanoparticles possess promising conductive properties, and they may transfer electrons, while carbon-based nanoparticles (graphene, graphene oxide/GO, or carbon nanotubes) possess electrochemical features. The connection between metal and carbon nanomaterials leads to a higher surface-to-volume ratio, which results in enhanced electrocatalytic properties [12].

Metal-based nanoparticles (e.g., silver, copper, copper oxide, zinc oxide) have already been defined as antibacterial with non-specific toxicity, which makes them able to fight pathogens that are listed as a priority [13]. This is especially important, given the new age of bacterial resistance, as many bacteria species have acquired resistance to bactericides and antibiotics, and many organic substances with antibacterial potential cause allergic reactions [14]. Despite this, graphene family materials may also exhibit antibacterial properties toward typical groups of pathogenic bacteria such as *Listeria monocytogenes*, *Staphylococcus aureus*, *Pseudomonas aeruginosa*, *Salmonella typhimurium*, *Streptococcus mutans*, *Escherichia coli*, *Staphylococcus epidermidis*, and *Enterococcus faecalis*, which may cause infections that are harmful to human health [15]. The planktonic bacteria model was the first to allow researchers to begin the work on antimicrobial agents. This enabled the production of biocides that could combat bacteria [16]. Even though most bacteria species are able to live in a biofilm structure, which is more complicated and different from that of planktonic cells [17], biofilm formation, nonetheless, starts with the adherence of free-floating planktonic bacteria cells to the surfaces [18]. Some of the genera (*Salmonella*, *Listeria*, *Staphylococcus*, *Clostridium*, *Campylobacter*, *Enterobacter*) are related to food products; therefore, they cause foodborne diseases because of the colonization of food products or the production of toxins which are subsequently ingested by humans [19,20,21]. Graphene family materials tend to agglomerate because of inner-plane interactions; thus, to counteract this process, the graphene-based materials’ surfaces have been modified and functionalized by a metal ions’, oxides’, and sulfides’ NPs or polymers [15]. In addition, the dispersion of some metal nanoparticles such as zinc oxide on the graphene sheets prevents their agglomeration by allowing this material to come into close contact with bacterial cells [22]. GO may constitute a functional platform for metal nanoparticles as it provides a long-term release of metal-based nanoparticles, even if more than two compounds’ nanocomposites are created [23]. However, the mechanisms of multicomponent nanocomposites have many more interaction points, given the effects that single nanoparticles may exhibit [24]. Given the many components that these composites may have, the creation of antibacterial nanocomposites is strongly needed to examine the strict mechanism of the antibacterial effect [25].

The aim of the study was to evaluate the mechanism of three-compound nanocomposites that can disrupt bacterial cells by the disorder of proton and electron flow. For this purpose, the following processes were carried out: for the physicochemical analysis of the nanomaterials, the zeta potential measurements were taken, a hydrodynamic diameter determination was made, a transmission electron microscope (TEM) visualization was conducted, and an FT-IR analysis was performed; to determine their impact on the bacteria cells, a viability analysis was performed via a PrestoBlue assay, and their ability to conduct colony formation, and the cell wall and membrane integrity were determined via a lactate dehydrogenase (LDH) assay, then, the bacterial conductivity and zeta potential, ATP concentration, and intracellular pH were measured. Thereafter, the toxicity toward human cells was verified.

## 2. Materials and Methods

### 2.1. Bacterial Culture

*Enterobacter cloacae* (ATCC BAA-2341), *Listeria monocytogenes* (ATCC 19111), *Salmonella enterica* (ATCC 13076), and *Staphylococcus aureus* (ATCC 25923) were obtained from LGC Standards (Teddington, GB) in the form of a spore suspension. Before their use in the experiments, the suspensions were defrosted and washed with distilled water to remove glycerol. The bacteria strains were cultured in nutrient media: tryptic soy agar (TSA) for *S. aureus* and *S. enterica,* and brain heart agar (BHA) for *L. monocytogenes* and *E. cloacae*. Before each experiment, the bacteria’s suspension in distilled buffer saline was prepared according to the appropriate density in the McFarland scale [26].

### 2.2. Physicochemical Analysis of Nanoparticles

Copper nanoparticle (Cu) (purity 99.8%, 25 nm), zinc oxide (ZnO), and graphene (GN) nanopowders (11–15 nm) were obtained from SkySpring Nanometerials (Houston, TX, USA), and GO was obtained from Advanced Graphene Products (Zielona Góra, Poland).

The zeta potential was measured by electrophoretic light scattering (ELS), and the hydrodynamic average was measured by dynamic light scattering using the Zeta Sizer Nano-ZS90 analyzer (Malvern Instruments, Malvern, UK) at room temperature for all nanomaterials, preceded by sonication at 500 W and 20 kHz for 2 min, and in case of nanocomposites, they were left for 15 min for self-combination.

TEM analysis was conducted with the use of the JEM-1220 (JEOL, Tokyo, Japan), operated at a voltage of 80 KeV. Droplets of each sample were placed onto TEM grids (Formvar on 3 mm 200 mesh Cu grids, Agar Scientific, Stansted, UK), and the samples were observed immediately.

FT-IR measurements were performed using a Nicolet iS10 (Thermo Scientific, Waltham, MA, USA) spectrometer. Before the sample measurement, a “dry air” background was recorded, which was subtracted automatically during the registration of spectra for the investigated samples. The samples were mixed with KBr at a ratio of 1/300 mg and then compressed at 7 MPa cm^−2^ to form a pellet, and the transmission spectrum was recorded. The spectra were collected in the range of 400–4000 cm^−1^.

The pH measurements of nanomaterials in ultra-pure water as a solvent were measured with the use of the pH meter SI Analytics HandyLab 100 (Xylem Water Solutions, Washington, WA, USA). The analysis was performed in triplicate at the room temperature.

### 2.3. Bacteria Viability

To analyze cell viability, the PrestoBlue assay (Cat. No. A13261, Invitrogen, Waltham, MA, USA) was conducted. The bacterial cultures were adjusted to 1.5 × 10^8^ cfu/mL in a distilled buffer saline; 90 μL of each suspension were added onto the 96-well plate, and 10 μL of ten concentrated nanomaterials (ZnO, 100 μg/mL; Cu, 25 μg/mL; GO, 10 μg/mL; and GN, 10 μg/mL, along with the following nanocomposites: GO, 10 μg/mL + ZnO, 100 μg/mL + Cu, 25 μg/mL; GN, 10 μg/mL + ZnO, 100 μg/mL + Cu, 25 μg/mL) were added into the appropriate wells and incubated for 24 h (37 °C). Thereafter, 10 μL of the PrestoBlue™ reagent was added, and after 10 min of incubation (37 °C), fluorescence was measured (λ_ex_ = 570 nm, λ_em_ = 600 nm) using a microplate reader (ELISA) (Infinite M200, Tecan, Durham, NC, USA). The assays were conducted for graphene and GO that was connected with zinc oxide and copper, separately, but the results from the three-component composites (the most effective for the most bacteria strains) were clearer and are thus, presented.

Cell viability was expressed as a percentage of control reduced by appropriate blank probes, where the control was the optical density of the wells with cells without nanomaterials and “blank” was the optical density of the wells with the medium without cells.

### 2.4. The Plate Count Method

To determine the number of colony forming units (CFU) that were affected by nanoparticle suspension, the plate count method was performed. The bacterial cultures were adjusted to 1.5 × 10^8^ cfu/mL in a distilled buffer saline and treated with nanoparticle suspensions (ZnO, 100 μg/mL; Cu, 25 μg/mL; GO, 10 μg/mL; GN, 10 μg/mL, along with the following nanocomposites: GO, 10 μg/mL + ZnO, 100 μg/mL + Cu, 25 μg/mL; GN, 10 μg/mL + ZnO, 100 μg/mL + Cu, 25 μg/mL). Samples were then incubated for 24 h under shaking conditions at 37 °C. Thereafter, a dilution of 10^−6^ was made and spread onto the nutrient agar plates (Biomaxima, Lublin, Poland). Plates were incubated for 24 h at 37 °C. The colony forming units were counted and transformed to the logarithmic scale.

### 2.5. Bacterial Cell Conductivity and Zeta Potential

The characterization of bacterial cell conductivity and zeta potential was assessed by ELS, with the use of the Zeta Sizer Nano-ZS90 analyzer (Malvern Instruments, Malvern, UK). The bacterial cultures (1.5 × 10^8^ cells/mL) were treated with nanoparticle suspensions (ZnO, 100 μg/mL; Cu, 25 μg/mL; GO, 10 μg/mL; GN, 10 μg/mL, along with the following nanocomposites: GO, 10 μg/mL + ZnO, 100 μg/mL + Cu, 25 μg/mL; GN, 10 μg/mL + ZnO, 100 μg/mL + Cu, 25 μg/mL). Conductivity and zeta potential were measured at room temperature immediately after the addition of nanoparticles and after 24 h of incubation (37 °C). The results were presented as the difference between the values obtained in 24 h and 0 h (the beginning of the experiment), reduced by a blank probe (nanosuspension without cells).

### 2.6. LDH Leakage

The membrane integrity was measured by the release of LDH using the Cytotoxicity Detection Kit (Cat. No. 11644793001, Sigma Aldrich, Hamburg, Germany); 90 μL of bacteria inocula (4.5 × 10^8^ cells/mL) were prepared in Mueller–Hinton broth (Biomaxima, Lublin, Poland) in a 96-well plate, and then 10 μL nanoparticle suspensions were added into each well to obtain the appropriate concentrations (ZnO, 100 μg/mL; Cu, 25 μg/mL; GO, 10 μg/mL; GN 10 μg/mL, along with the following nanocomposites: GO, 10 μg/mL + ZnO, 100 μg/mL + Cu, 25 μg/mL; GN 10 μg/mL + ZnO 100 μg/mL + Cu 25 μg/mL). The bacteria suspensions without nanoparticles were used as the control, and the nanoparticles without bacteria suspensions were the blank probes. After replenishing each well, the microplate was centrifuged at 3000 rpm for 5 min. One hundred microliters of the supernatant were transferred to 96-well plates, and 100 μL of the LDH assay mixture were added to each well. The plate was incubated for 30 min at room temperature. The optical density of each well was recorded at 450 nm on an ELISA reader (Infinite M200, Tecan, Männedorf, Switzerland). LDH leakage was expressed as the percentage between the optical densities of the samples tested and of the control probes, both reduced by the optical density of the blank probes for appropriate samples.

### 2.7. ATP Concentration

ATP concentration was measured using the ATP Assay Kit (Cat. No. MAK190, Sigma Aldrich, Hamburg, Germany). The bacteria inocula (4.5 × 10^8^ cells/mL) were treated with the appropriate nanosuspension concentrations for 24 h (37 °C), and the samples were placed in a 96-well plate with a volume of 50 μL; 50 μL of the reaction mix was added into the appropriate wells. The standard curve and preparation of the samples were conducted according to the manufacturer’s instructions. The fluorescence (λ_ex_ = 535/λ_em_ = 587 nm) was measured after 30 min of incubation of the 96-well plate that was protected from the light in a microplate reader (Infinite M200, Tecan, Männedorf, Switzerland).

### 2.8. Intracellular pH

To examine the intracellular pH of the cells, the Fluorometric Intracellular pH Assay Kit (cat. no. MAK150, Sigma Aldrich, Hamburg, Germany), using BCFL-AM, was used. The bacteria were centrifuged for 5 min at 5000× *g*, the pellet was suspended in PBS (McFarland scale, 0.5), and 100 μL of the dye-loading solution was added into each well. The cells were incubated, while they were protected from the light, for 30 min at 37 °C, followed by incubation at room temperature for an additional 30 min. The compounds that were tested (nanoparticles and nanocomposites) were added into each appropriate well with a volume of 50 μL. The fluorescence (λ_ex_ = 490/λ_em_ = 535 nm) was measured after 3 min from the addition of compounds using a microplate reader (Infinite M200, Tecan, Männedorf, Switzerland).

### 2.9. Cytokine Antibody Array

The cytokine expression profiles in HFFF-2 (ATCC, Manassas, VA, USA) were detected using Human Cytokine Antibody Array Membrane (Cat. No. ab133997, Abcam, Cambridge, UK) for 42 targets. HFFF-2 was maintained in Dulbecco’s modified Eagle’s culture medium, containing 10% fetal bovine serum (Life Technologies, Houston, TX, USA), with 1% penicillin and streptomycin (Life Technologies) at 37 °C in a humidified atmosphere of 5% CO_2_/95% air in the NuAire DH AutoFlow CO_2_ Air-Jacketed Incubator (Plymouth, MN, USA). When the cells reached 80% confluence, the medium was removed, and the nanosuspensions were added into the cell cultures and incubated for 24 h. Thereafter, the cells were trypsinized and harvested. The protein extracts were prepared using TissueLyser LT (Qiagen, Hilden, Germany) by centrifugation at 13,000× *g* for 10 min at 4 °C, with a pre-frosted adapter at 50 Hz for 10 min. The concentration of protein extracts was determined using the BCA Protein Assay (Thermo Scientific). Three samples from each group were diluted to a final concentration of 5 μg/μL. The procedure was performed according to the manufacturer’s instructions. The immunoblot pictures were captured using Azure Biosystem C200 (Dublin, CA, USA) and analyzed with the Protein-Array Analyzer plugin of the ImageJ software (version 1.50e, National Institutes of Health, USA).

### 2.10. Statistical Analysis

All the data are represented as mean values with standard deviation. One-way analysis of variance with the post-hoc Tukey test (HSD) was performed using the GraphPad Prism 9 software (version 9.2.0, San Diego, CA, USA). The statistically significant differences were considered at *p*-value ≤ 0.05.

## 3. Results

### 3.1. Physicochemical Analysis of Nanoparticles

The nanoparticles that were tested were able to create agglomerates, which is evident, given the values of the zeta potential (Figure 1 and Table 1) and the TEM analysis (Figure 2). Interestingly, ZnO and GNZnOCu had positive zeta values (2.32 mV and 10.56 mV, respectively) in comparison to the other samples, which all had negative values. The highest values were observed in the GO (−27.53 mV) and GN (−27.40 mV) samples, and they were also the most stable samples, reaching zeta values closest to the limit value of ±30 mV.

The mean hydrodynamic diameter indicates that all the samples exceeded the nanometer scale. However, as shown in the TEM analysis, the large agglomerates contain much smaller nanoparticles, which is due to their instability when they are clumped together. It is also shown in the mean average that was gained by DLS (Figure 1), that GN, GO, and GNZnOCu had two fractions of particles in which one was much bigger than the other one. Furthermore, GN and GO were in the form of flakes; thus, their hydrodynamic diameters were bigger than those of the metal nanoparticles. The same phenomenon was observed in the case of nanocomposites, where GN and GO constituted the platform for other nanoparticles.

The pH of the nanomaterials tested were: 6.51 (±0.12) (water), 6.55 (±0.07) (GN), 6.49 (±0.1) (GO), 6.76 (±0.06) (ZnO), 6.59 (±0.03) (Cu), 6.63 (±0.05) (GNZnOCu), and 6.50 (±0.05) (GOZnOCu).

The FT-IR spectra of GN, GO, and ZnO as well as their composites are shown in Figure 3. The broad peak observed between 3000 and 3650 cm^−1^ is assigned mainly to the water and hydroxyl groups (O-H). The smaller features from 2850 to about 3000 cm^−1^ can be attributed to the C–H stretch and C–H bending around 1310 cm^−1^. The peak around 1615 cm^−1^ can be assigned to aromatic (sp^2^ vibrational) C=C bonds that are present in graphitic carbon. Another peak was observed around 1500 cm^−1^ on the FT-IR spectrum revealed C=O bonds. An absorption band at 1385 cm^–1^ was present because of aliphatic C–C bonds (sp^3^ C–H bend). The overlapping peaks, which form an absorption band in the 1300–950 cm^−1^ region, can be attributed to C–O moieties existing in a different structural environment [27,28].

### 3.2. Bacteria Viability

In all the bacterial species, one of the types of the nanocomposites were the most toxic compounds, and bare GN was less toxic than these (Figure 4). GNZnOCu was the most toxic compound for *L. monocytogenes* and *S. aureus,* while GOZnOCu was the most toxic compound for *E. cloacae* and *S. enterica.* GO caused a viability of approximately 80% (ZnO caused slightly less). Cu were harmful compounds to a similar extent to nanocomposites (more harmful for *E. cloacae* than GNZnOCu, and for *L. monocytogenes* than GOZnOCu). In Gram-negative species, GOZnOCu was more toxic than GNZnOCu, which was contrary to Gram-positive species, where the latter caused a higher decrease in viability.

### 3.3. The Plate Count Method

One of the types of nanocomposites were the most limiting factor of all the probes (Figure 5); these were: GOZnOCu for *E. cloacae* and *S. enterica*, and GNZnOCu for *L. monocytogenes* and both for *S. aureus*. Metal nanoparticles were effective toward most bacteria species, especially for *L. monocytogenes* (in a similar extent as GNZnOCu). GN and GO did not limited colony formation, however a slight decrease was observed in the GO probe for *S. aureus* and GN for *L. monocytogenes.*

### 3.4. Bacterial Cells’ Conductivity and Zeta Potential

The zeta potential values of Gram-positive bacteria were positive in contrast to those of Gram-negative bacteria, which were less than zero (Figure 6). The factor that was most different from the control in the case of *L. monocytogenes* and *S. aureus* was GNZnOCu, while for *E. cloacae* and *S. enterica*, these were GO and GOZnOCu. The values for GOZnOCu in the Gram-positive bacteria decreased, but in the Gram-negative bacteria, they increased. GNZnOCu and ZnO, in control, exhibited a positive zeta potential.

### 3.5. LDH Leakage

The largest amount of LDH release was observed in one of the types of nanocomposite probes, among all the bacteria that was tested (Figure 7). For *E. cloacae* and *S. aureus,* the most toxic factor was GOZnOCu, as well as for *S. enterica,* but in a similar extent as GO. GNZnOCu was the most reactive for *L. monocytogenes*. Cu was more reactive than GNZnOCu for *E. cloacae* and it was also more reactive than GOZnOCu for *L. monocytogenes.* GN and GO caused slightly higher values of LDH release in comparison to the negative control. A higher LDH release was observed in the Gram-positive species, in which, after treatment with nanocomposites, these values were closer to the positive control, which indicates the maximum amount of LDH leakage.

### 3.6. ATP Concentration

In all the bacteria species, the nanocomposites caused the smallest ATP concentration among all the samples, and the highest concentration in the GN and GO probes (Figure 8). ZnO also caused decreases in the ATP concentration in all the species; however, a significant decrease in the case of Cu was observed only in the Gram-positive species. Based on the results obtained, the metal nanoparticles led to a greater reduction of ATP than the graphene-based nanoparticles did. Additionally, GOZnOCu was the most limiting factor, with a stronger effect than GNZnOCu.

### 3.7. Intracellular pH

The intracellular pH assay measured decreases in the cells treated with various substances (Figure 9). Interestingly, the decrease in fluorescence was observed in all the bacteria species (except *E. cloacae*) that were treated with GOZnOCu. This decrease was also observed in the Gram-positive species (*L. monocytogenes* and *S. aureus*) in probes with GN and GO, which does not occur in the case of the Gram-negative species.

### 3.8. Cytokine Antibody Array

A slight increase in the protein expression was observed for TNF-β, growth-related oncogene (GRO), and thymus and activation-regulated chemokine (TARC) for HFFF-2 cells that were treated with GNZnOCu (Figure 10). The increase in the level of GRO was observed for GNZnOCu. The increase in TARC was observed for Cu and GNZNoCu. TNF-β was observed, to a small extent, in GOZnOCu, and at a higher level for the Cu and GNZnOCu samples.

## 4. Discussion

The obtained results indicate that it is possible to create three-compound nanocomposites that exhibit great antibacterial properties. However, given the many aspects influencing the behavior of nanocomposites, it is especially important to investigate them in depth.

The shape of the nanoparticles was similar to those in other studies, where graphene sheets and GO composed thin flat flakes [29]. Because of their shape, they appeared to be large when analyzed by DLS which is based on light scattering, and which is calculated using the Stokes–Einstein equation, and one of its elements is the hydrodynamic diameter of an equivalent spherical particle [30]. For the same reason, the nanocomposites appeared large. However, the metal nanoparticles, Cu and ZnO, had the diameters of aggregates that were 682 and 2155 nm, respectively, but both types were unstable, which was visible in the zeta potential measurements (Cu = −19.03 ± 0.51 mV; ZnO = 2.32 ± 0.83 mV). None of the values that were obtained exceeded ±30 mV, which would provide them with a strongly cationic or anionic character [31]. Therefore, they are large as a result of us creating agglomerates by them. This, in turn, influences the biological effect that the nanoparticles will have [32]. The analysis of the zeta potential in a different pH indicated (Appendix A) that the zeta potential became more negative as the pH increased. In our results, the nanosuspensions had pH values of around 6.5. Due to that aspect, they may be able to agglomerate, especially as bacterial culture additionally acidifies the environment [33], so in the tests that were performed, nanoparticles may have had the tendency to agglomerate. In addition, the agglomeration of the nanoparticles was clearly visualized in the TEM analysis (Figure 2). As mentioned before, the creation of aggregates/agglomerates influences the biological effect that is induced by the nanoparticles. It should be remembered that research on the influence of pH on nanoparticle stability and agglomeration state reflects the situation in an aqueous solution. There will be different situations in both in vitro and in vivo studies, in which the conditions will be additionally changed due to the presence of proteins, lipids, and other biomaterials that may adhere to the nanoparticle surface and change its properties, chemistry, and agglomeration state, and this will have a serious impact on the toxicology aspects [34]. Cu aggregation may be the result of rapid oxidization under the conditions of exposure to the atmosphere [35]. Given the preparation of the nanoparticles, which were in normal conditions (with room temperature and had access to oxygen), Cu, despite having a fairly negative zeta potential, may create agglomerates/aggregates exceeding 600 nm. ZnO in the TEM analysis showed aggregates with single nanoparticles that exhibited different shapes and sizes. Similar results for the visualization of ZnO were obtained by Mendes et al. [36]. In general, ZnO has a positive surface charge, which is compatible with our results, so it can be easily connected with negatively charged structures [37]. The FT-IR analysis showed that the bonds presented in the components, alone, are compatible with those found in the composites (Figure 3). Given the presence of C=O groups in the nanomaterials, the bond between the nanocomposites and the bacterial cell surface (amines of peptidoglycan and amino acids) can have a strong character [38].

Bacteria exposure to antimicrobials results in two basic mechanisms: bacterial injury or agent adhesion to the bacterial surface, and both of these mechanisms occur simultaneously [39]. In our results, the latter occurred. GO and GN, which were both in the form of flakes, were not highly effective antibacterial factors, which was visible from the PrestoBlue assay. These nanomaterials, applied in the concentration of 10 µg/mL, limited the viability to a small extent. PrestoBlue is a good indicator for bacterial growth and viability, despite the fact that some of the light sensitivity and test time is dependent on the bacterial metabolism [40]. Nevertheless, there are some reports, described previously, that state that carbon nanoparticles can cause the re-oxidation of resorufin (pink) to resazurin (blue) or hyper-reduction of resorufin (pink) to hydroresorufin (colorless), which affects the intensity of the color obtained in the test [41]. GN and GO, given their physicochemical analysis (Figure 1 and Figure 2), were multilayered, and so their weak antibacterial action is caused by their form because single flakes have a stronger antibacterial effect [42]. All the samples—even GN and GO, which were slightly toxic for bacteria—exceeded the LDH leakage levels in relation to the negative control; however, this exceedance was not high in all the species. Remarkably, the conductivity of bacteria cells that were treated with GN and GO increased in all the cases, relative to the control. In contrast, the ATP assay indicated that these nanomaterials did not cause the limitation of the ATP content, but the internal pH was altered. Interestingly, these types of nanomaterials have the most negative zeta values in the physicochemical analysis, while in the analysis of bacterial zeta potential, the samples that were treated with them had values of greater than 0. An increase in the values of the zeta potential indicates strong binding between the cell surface and the dispersing solution [43]. The pH value, which is one of factors that influences the zeta potential, in the case of the nanomaterials in the medium was around 7.3, and this did not change over time (Appendix A). However, bacteria, during the growth process, acidify the environment [33], which may level out the zeta potential values (Appendix A). Given the abovementioned facts, GO and GN may adhere to the bacterial surface and wrap around it, separating them from the surrounding medium and the nutrient, which is one of the already known graphene material antibacterial mechanisms [44], but this does not disrupting the cells mechanically, so cell death has not happened to that extent. An increase in the GN and GO samples was therefore observed in the colony forming units (Figure 5), because the bacteria cells were wrapped by the graphene-material sheets and during the spread plate method, the bacteria adhered with their surface to the surface of the agar plates, and these could grow easily if they were not disturbed mechanically. The mechanism wherein GO sheets wrap around the bacteria cells was shown in our previous study (Lange et al., 2022), where bacteria were directly on the surface of the GO flakes [45]. In the other tests that were presented, where the bacteria were grown in a liquid media, the flakes may have isolated them from the surrounding medium and thereby, affected their metabolism, causing effects such as a disturbance of conductivity.

A different pattern is observed with the metal nanoparticles, which may exhibit many mechanisms, including membrane disruption, ATP depletion, and ion release [46]. The metal-based nanoparticles interfere with the cell surface as this is the first interaction point, which generates the disruption of the cell membrane potential and integrity because of the process of electrostatic binding. By breaking the cell membrane, a large amount of cytosol is released, and the bacteria try to prevent this through proton efflux pumps and electron transport [47]. Considering the results obtained, Cu and ZnO decrease the bacterial viability (Figure 4), contributing to LDH leakage and a substantially reduced ATP. Metal nanoparticles influence the surface charge of the bacteria and affect their membranes (Figure 7), especially after the ZnO treatment. Also, for the ATP analysis, ZnO was the one that limited the ATP concentration in all bacteria species (Figure 8). A high concentration of ATP supports the bacteria division processes [48], thus, metal nanoparticles may be inhibited by the ability of bacteria to multiply, which resulted in the colony formation disturbances, to a small extent (Figure 5). The reduction of about 3 log CFU/mL after the exposure to Cu and ZnO was observed only for *L. monocytogenes*. A reduced number of *L. monocytogenes* by 1 log CFU/mL was reported in the study by Skowron et al., when they were treated with copper and silver nanoparticles, thus causing a reduction of 90% [49]. Cu, as a nanocolloid, had the value of the zeta potential equal to −19.03 mV, but the bacterial cells after treatment with those nanoparticles reached less negative values, which also surpassed the control values in three strains: *E. cloacae*, *L. monocytogenes*, and *S. enterica*. Only *S. aureus* had more negative values than its control. This suggests that despite the limited viability, bacteria may multiply, increasing the negative charge that they exhibit on the surface, especially since the LDH assay clearly showed that Cu was the least toxic compound and the cell contents did not leak, which could contribute to an increase in the zeta potential value. Analyzing ZnO, which had positive zeta potential (2.32 mV), in Gram-negative species, the values were more neutral (closer to 0) after the exposure of positively charged nanoparticles. Similar results were obtained by Halbus et al., where the reduction of the negative zeta potential value was observed after the binding with positively charged nanoparticles [50]. However, the antibacterial activity of ZnO is inversely proportional to its size [51]. In our study, ZnO was large (hydrodynamic diameter of 2155.33 nm), with a tendency to agglomerate, so its mode of action might be limited. In the viability test, ZnO was slightly toxic to bacteria cells, which was similar to the LDH assay, but the ATP content was considerably reduced. This may be the cause for the surface charge because opposite charges bind to each other more easily, and ZnO had a positive charge (2.3 mV), whereas both the Gram-positive and Gram-negative bacteria exhibited negative charges on the surface because of the presence of lipopolysaccharides in the Gram-negative bacteria and teichoic acids in the Gram-positive bacteria [52].

The situation is different at the stage of nanocomposite usage. Their mechanism of action is multilevel. As previously shown, nanocomposites have a stronger antibacterial effect than their components do, alone [45,53]. An addition of already known nanoparticles with antibacterial properties to other carriers provides support for the nanoparticles and enhances the antimicrobial activity, thereby making wider use for them in biomedical fields a possibility [54]. As shown in a TEM analysis, GN and GO provide platforms for metal nanoparticles due to their flake’s form. Nanocomposites may disrupt the cellular outer membrane and wall, but they may cause different disturbances in cell metabolism and life processes, and such a phenomenon was observed in our study. First, nanocomposites cause the biggest limitation to viability as well as LDH release and ATP reduction. Considering that fact, the bacterial cells formed fewer colonies after the treatment with nanocomposites (Figure 5) than they did with the metal nanoparticles. Differences in viability (Figure 4) and CFU number (Figure 5) may be the result of a reaction between carbon nanoparticles (GN and GO) with resazurin [41], but the general trend that nanocomposites were the most toxic was upheld in both cases. The metal nanoparticles damaged the bacteria mechanically through direct contact, as GN and GO constitute a platform which separate them from environment. It is believed that GO may entrap bacteria cells, which inhibits their division ability [25]. The enhanced antibacterial activity of nanocomposites was observed by Fontecha-Umaña et al., where they noted that the effect was more than 2 log CFU/mL because of the stabilization and slower release of metal ions [55]. The phenomenon of ion release affecting the cells may also have existed in our research, especially as the plate count method did not show the mechanism of the cellular effect, and it may pose a challenge when there are viable but not culturable cells, which have intact membranes and genetic material, but have different metabolic functions from viable culturable cells [56]. For this reason, we also analyzed other aspects of bacterial cell function in order to illustrate the possible action of the complex structures such as three-compound nanocomposites. LDH leakage, which may occur while the external layer is interrupted, was visible after the nanocomposites’ treatment, to the largest extent (Figure 6). The limitation of CFU in the nanocomposite probes may be the results of membrane disturbances (Figure 7) and a reduction of ATP content (Figure 8), which resulted in the decreased ability to perform cell division and thus, resulted in a lower number of colonies being formed. However, their bacteria zeta potential is not consistent (same as the conductivity), which may be because the nanocomposites consisted of many compounds, among which each one has its own physicochemical properties. One study suggested that the GO-metal substrate changed the electron transfer process by absorbing electrons from the bacteria respiratory pathways and modifying the oxygen-containing functional groups on the GO surfaces [57]. Such a prediction may be true as the nanocomposites have significantly altered the conductivity of the bacteria, which may suggest the occurrence of the abovementioned phenomenon. Furthermore, a nonconductive substrate such as the GO/Glass system had no antibacterial activity [58], which suggests that metal nanoparticles, in addition to the graphene-based nanomaterials, changed their ability to disrupt the conductivity and surface potential.

Apart from the materials, the structure of the bacteria is extremely important because their main interaction point with any antibacterial agent is the external surface [4]. Various structures occur on the bacterial surface, which differ depending on the cell wall (Gram-positive and Gram-negative). Teichoic acid (Gram-positive) or lipopolysaccharides and phospholipids (Gram-negative) are associated with both basic and acid functional groups. Such connections determine the bacteria’s electrostatic behavior, especially in regard to its interaction with various agents [59]. Plasma membranes are multifunctional in both Gram-positive and Gram-negative bacteria; they constitute the site of active transport and respiratory chain components as well as the energy-transducing systems and the H+-ATPase of the proton pump [60]. Therefore, in our study, if the integrity of the membrane was disrupted, the ATP content, conductivity, and zeta potential of the bacterial cells would also be altered, along with the intracellular pH, since all these functionalities depend on the proper functioning of the membrane.

In this research, the Gram-positive species demonstrated a lower susceptibility than the Gram-negative species did, based on the viability and integrity assays (Figure 4 and Figure 7). However, the results that were obtained did not exhibit an unambiguous tendency as some species are more resistant than others. The cell wall structure of the Gram-positive bacteria is difficult to penetrate because of its thickness, and nanoparticles act only on the bacteria surface through electrostatic binding. This causes membrane depolarization and changes the potential, which results in the loss of integrity that finally leads to the interruption of energy transduction and cell death [61]. In this study, the bacteria had changed the electrostatic potential, which was differentiated by the group (Figure 6). The zeta potential values, in terms of bacterial function and metabolism, are strain dependent. However, this parameter may be used in accessing the interaction between the bacteria and the antibacterial agent, the dysfunction of the cell membrane, and ultimately in bacterial viability [39]. The zeta potential and conductivity results may be understood in two ways. Highly negative values of zeta potential may indicate that the bacteria will multiply, thus, increasing its negative charge. However, in live cells, ion homeostasis is maintained because of the functional transport systems (such as efflux pumps). In dead cells, this system is inactive; thus, ions may leak from cells that are remaining on their surface and change their charge to negative [43]. Additionally, the electrical conductivity may be changed by the metabolic processes during microbial growth, given the presence of the ions that are generated throughout those processes [62]. This suggests that the obtained results are not unambiguous, making the viability analysis (Figure 4) and ATP analysis (Figure 8) worth considering. However, when some structures get inside, bacteria may overcome the risk from antimicrobial agents such as nanoparticles by the energy-dependent active efflux of toxic ions [61]. This means that effective structures can have a weaker effect once the bacteria are able to “pump out” the threatening compounds. Such a phenomenon may be present in the case of *S. aureus* that was treated with GOZnOCu, which had the lowest zeta potential value in comparison to the control (Figure 6), but in other tests, the composite was the most toxic agent. This theory confirms the results of the pH assay, in which the decrease in fluorescence intensity was the lowest, but the intracellular pH was also influenced by the movement of the efflux pumps, which depends on the ATP. When the cytoplasmatic pH varies, it contributes to power or inhibit the protonmotive force (PMF), which is the electrochemical or chemiosmotic gradient of protons, determined by active proton pumping and secondary ion movements [63]. BCECF/AM may also be used to label clinical isolates to check the uptake [64]. Nonfluorescent BCECF/AM esters are hydrolyzed to a fluorescent BCECF acid form by an intracellular esterase, after entering the cell [65,66]. The fluorescence intensity of the intracellular BCECF may provide information about the intracellular pH [67], as the acid form of the dye is maintained inside the cell. However, acquiring such measurements may be complicated in the case of bacteria because some bacteria may excrete BCECF from their cells as a result of the ATP-driven efflux pumps [67]. Therefore, the samples that did not have a depleted ATP content could successfully pump out the BCECF of the cells, therefore giving disrupting results. Such a phenomenon could be the case for *E. cloacae* and *L. monocytognes*, which had drastically depleted ATP levels when they were treated with nanocomposites, and an increase in fluorescence was observed. Under standard growth conditions, most bacteria have a higher internal pH than in the surrounding environment does. The resulting pH gradient can be used as a motor force for processes requiring the supply of energy, such as ATP synthesis [68]. Not surprisingly, in our results, the differences in fluorescence in the measurement of intracellular pH appeared in the samples where the contents of ATP were disturbed. PMF that is generated across the cell membrane is based on the extrusion of protons by the electron transport chain. This is necessary for ATP synthesis in bacteria cells. PMF also constitutes the crucial element of bacterial survival, but in antimicrobial therapy, it has been largely omitted for concerns of toxicity, which could have similar effects on other cells [69].

One of the tested bacteria showed results with higher values than the other species in conductivity, zeta potential, and intracellular pH. *Listeria monocytogenes* is a Gram-positive pathogen, but its structure is not typical for Gram-positive bacteria and it is similar to that of Gram-negative bacteria because of the presence of partially deacetylated N-acetyloglucosamine residues and the ability for it to encode a high number of surface proteins [70]. There are suspicions that *Listeria monocytogenes* is the only Gram-positive bacteria that contains authentic LPS [60]; however, other studies have disproved this idea [71]. Furthermore, this bacteria species is characterized by small cells, around 0.5–4 μm in diameter and 0.5–2 μm in length [72]. Given its ability to invade and live inside cells, *Listeria* possesses defense factors that allow it to modify its metabolism, enabling it to survive. This main mechanism is based on the action of alternative sigma transcription factors, which are activated during environmental stress, nutrient deficiency, or altered pH. The nanoparticles/nanocomposites that are introduced into the bacterial suspension affected all these environmental factors, so it is very likely that this bacterial species coped with them through the action of sigma transcription factors. However, these factors are present in Gram-positive species, and also in *S. aureus*, although in our study, this bacterium had greater wall and membrane damage, which was the main cause of bacterial death, so the transcription of these factors may not have occurred. Furthermore, it has been shown that *Listeria* manages pH homeostasis differently from the way other species do [73], which, in the case of this study, may explain the high values that were obtained.

To summarize, the probable mechanism of action of graphene-based nanocomposites that are decorated with ZnO and Cu is multidimensional (Figure 11). GN and GO, which serve as a platform for metal nanoparticles, wrap around the bacteria and separate them from the medium containing the nutrient. When the structures are close to the cells, at the same time, with the addition of metal nanoparticles, ions are generated. They cause the modification of the zeta surface potential, membrane interruption, and LDH leakage, which cause cell death. Membrane integrity alteration and separation from the medium cause changes in the bacterial metabolism, which subsequently causes disorders of surface charge and ATP content depletion because of the alteration of proton and electron flow.

Nanomaterials may constitute an alternative for traditional antibiotics, owing to more specific targeting and controllability. Clinical applications of nanoparticles, however, are not without their difficulties, which are related to the delivery method of the therapy [74]. Novel methods of applying nanoparticles, such as by solid-film laser transfer, are currently in use [75], although basic research regarding the properties of colloidal nanoparticles and their interaction with bacterial cells must be carried out in order to be able to introduce new therapeutic solutions. The toxicity of nanoparticles when they are used is a crucial element in further applications to animals and humans. Many aspects influence their interaction with cells and tissues, owing to the unique properties of nanoparticles. Its effect varies depending on type, dose, physicochemical property, and the type of cell line that is used. Given that there are several aspects that need to be considered, the specific impact is still not fully understood [76]. The possibility of toxicity toward human cells is present in Figure 10. The assay enabled the detection of 42 cytokines at the same time. Based on our results, only three types of cytokines have been identified: TNF-β, GRO, and TARC. While toxicity was not significant, given the possibility of harmful effects, nanomaterials should be analyzed in depth.

TNF-β, which is a tumor necrosis factor, is also known as lymphotoxin. It is involved in cell proliferation, differentiation, and apoptosis, and it is connected to the cell permeability membrane. It is toxic toward many cancer cells [77]. An increase in the level of TNF- β was observed in the cases of CuNPs, GNZnOCu, and GOZnOCu treatment; however, in Cu and GNZnOCu, the increase was the highest of all the samples. TARC, also known as chemokine ligand 17 (CCL17), acts as a chemoattractant of T-helper-2 (Th2) cells. This chemokine is connected with some respiratory diseases, including bronchial asthma, allergic rhinitis, and eosinophilic pneumonia [78]. The increase in TARC expression was observed in the HFFF-2 cells that were exposed to Cu and GNZnOCu. GRO is a member of the chemokine family that plays a critical role in inflammatory reactions and wound-healing processes, but it is also related to tumorigenesis, angiogenesis, and metastasis [79]. GRO was observed in the cells after their treatment with Cu, only. The obtained results suggest that Cu, when incorporated into the composite, shows a less toxic effect toward the HFFF-2 cell line. Other components of nanocomposites (GN, GO, or ZnO) were not toxic toward HFFF-2 cells and did not cause a rise in the cytokine levels.

## 5. Conclusions

Each type of bacteria reacts differently and has its own defense factors against unfavorable environmental conditions; on the other hand, each type of nanoparticle has a unique effect. For this reason, we have demonstrated the actions of nanocomposites consisting of three components, which evidently exhibit a multifaceted effect, damaging bacterial cells through different mechanisms of action, which include the mechanical damage of the membrane, changes in conductivity and surface charge, ATP content reduction, and changes in the intracellular pH. Despite the nanocomposites’ excellent antimicrobial properties, general toxicity must always be considered during their possible application, which often needs to be further investigated to ensure safety.

## Figures and Tables

**Figure 1 nanomaterials-12-03058-f001:**
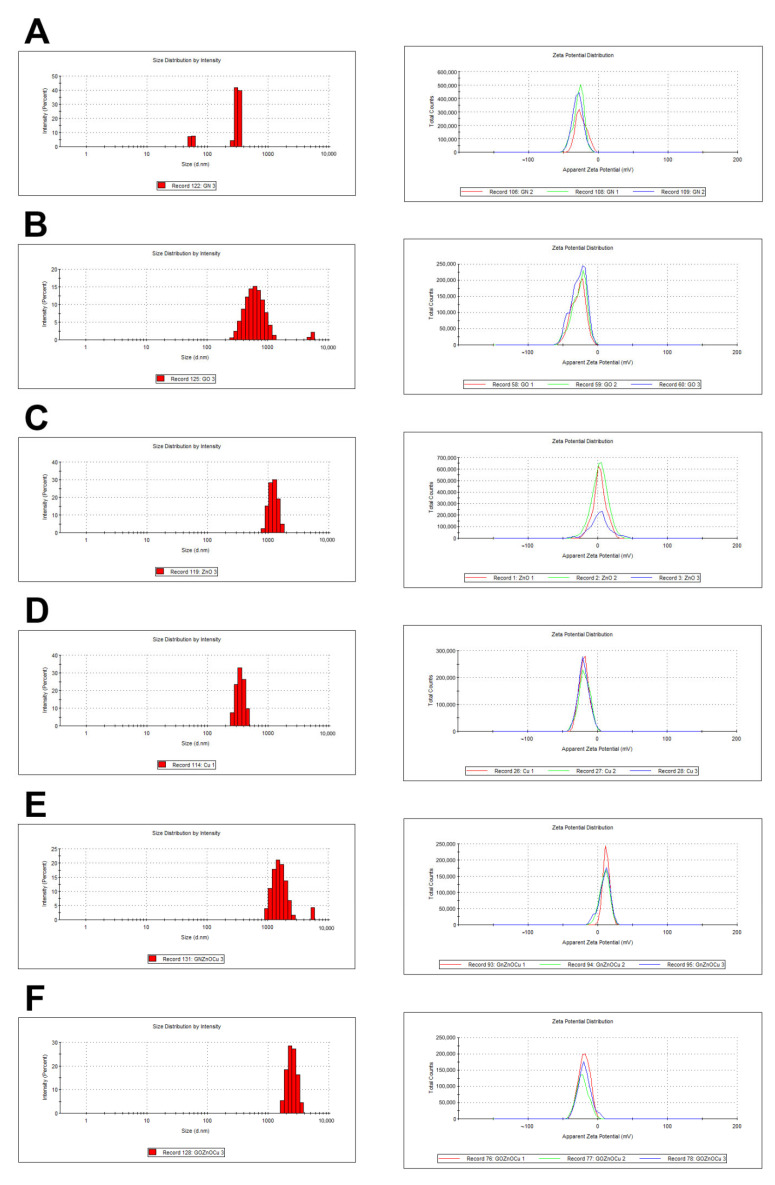
Physicochemical analysis (mean average and zeta potential, respectively) of nanoparticles: (**A**) GN; (**B**) GO; (**C**) ZnO; (**D**) Cu; (**E**) GNZnOCu; (**F**) GOZnOCu. The mean average was measured by DLS, and the zeta potential was measured by ELS. Three different colors mean repetitions.

**Figure 2 nanomaterials-12-03058-f002:**
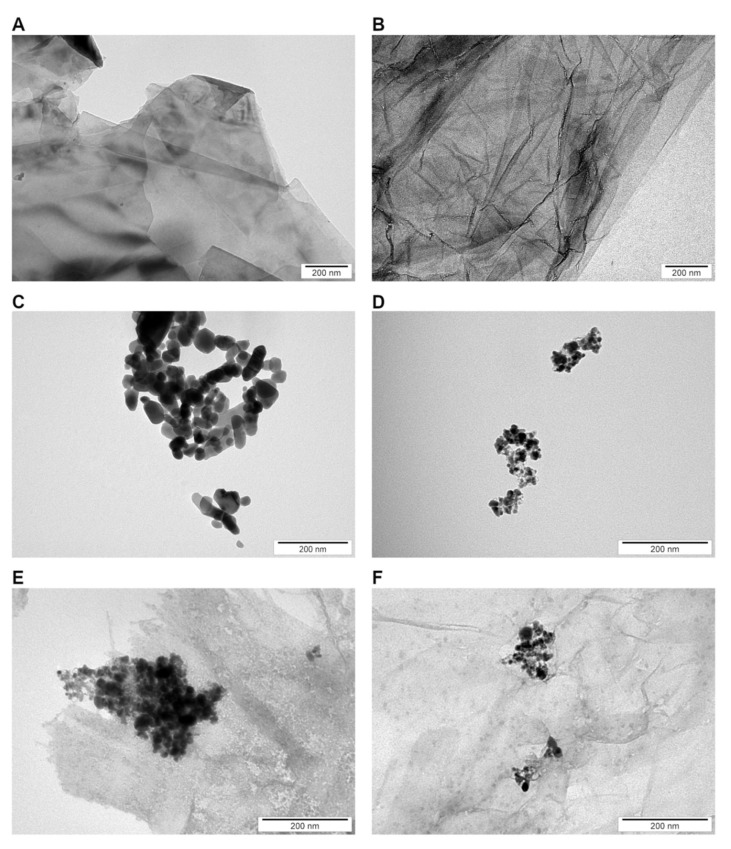
Visualization of nanomaterials by TEM: (**A**) GN; (**B**) GO; (**C**) ZnO; (**D**) Cu; (**E**) GNZnOCu; (**F**) GOZnOCu.

**Figure 3 nanomaterials-12-03058-f003:**
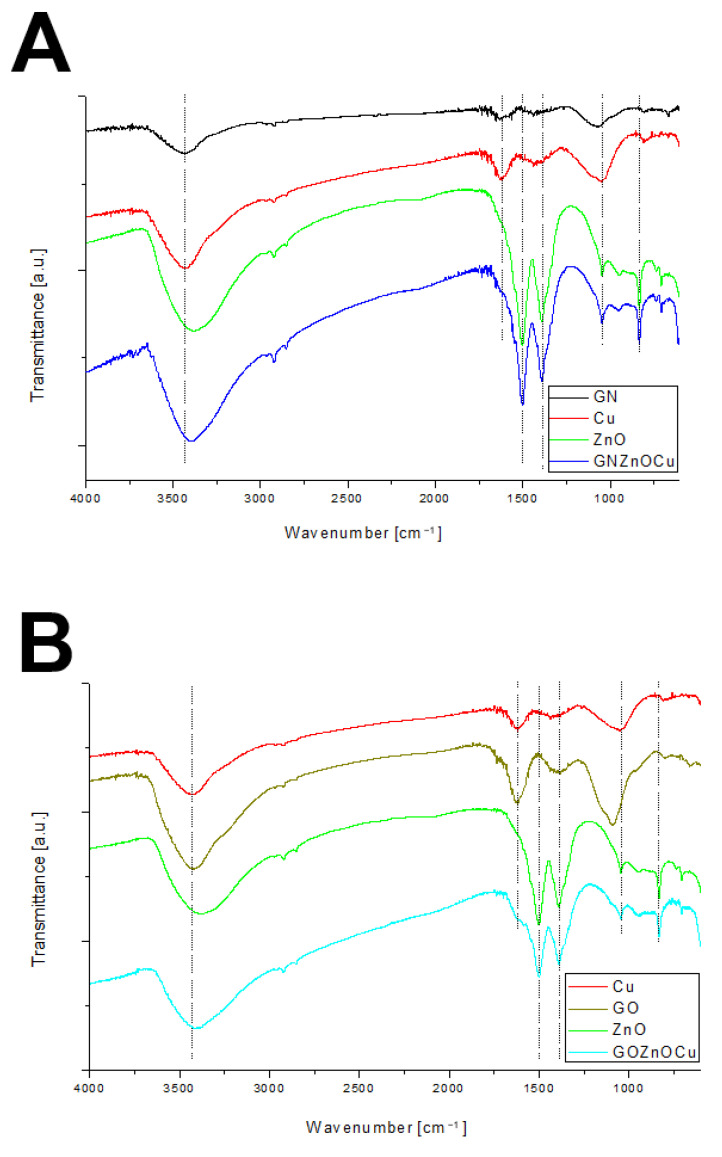
FT-IR spectra of nanomaterials: (**A**) GNZnOCu nanocomposite and its components; (**B**) GOZnOCu and its components.

**Figure 4 nanomaterials-12-03058-f004:**
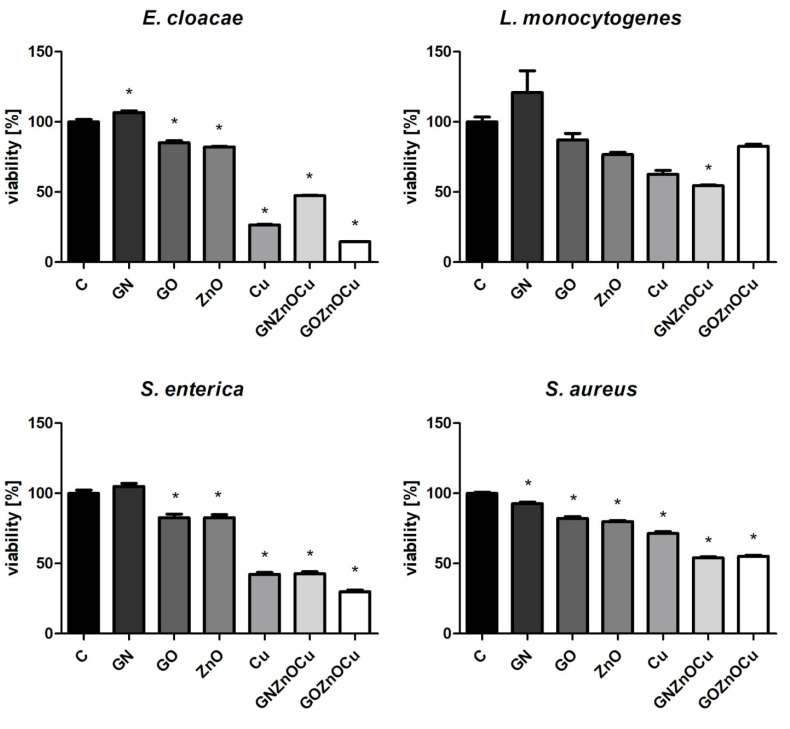
Viability of bacteria after treatment with nanoparticles. C is the control sample, and the results are mean values ± standard deviation. “*” symbols indicate statistically significant differences in comparison to the control.

**Figure 5 nanomaterials-12-03058-f005:**
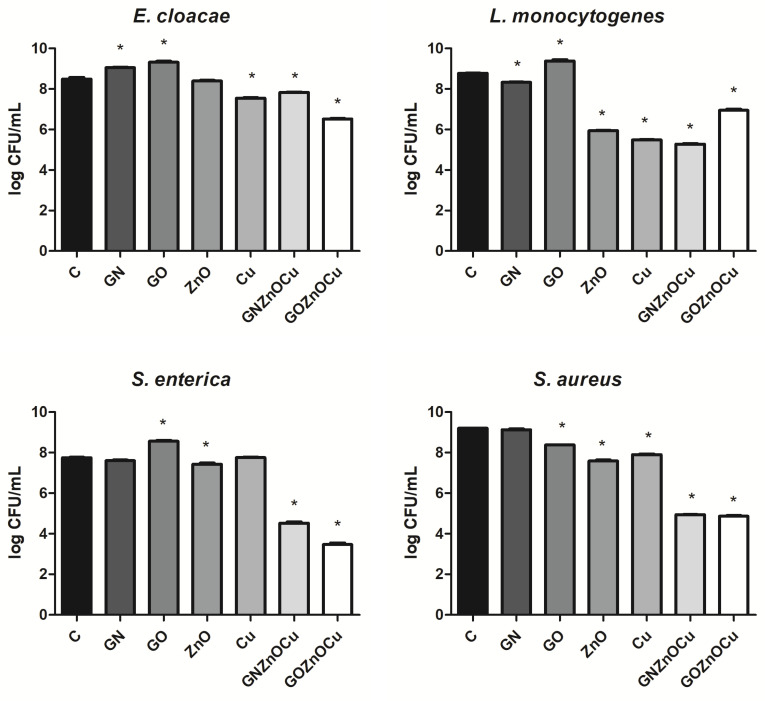
Mean CFU/mL (log) for bacteria after treatment with nanoparticles. C is the control sample, and the results are mean values ± standard deviation. “*” symbols indicate statistically significant differences in comparison to the control.

**Figure 6 nanomaterials-12-03058-f006:**
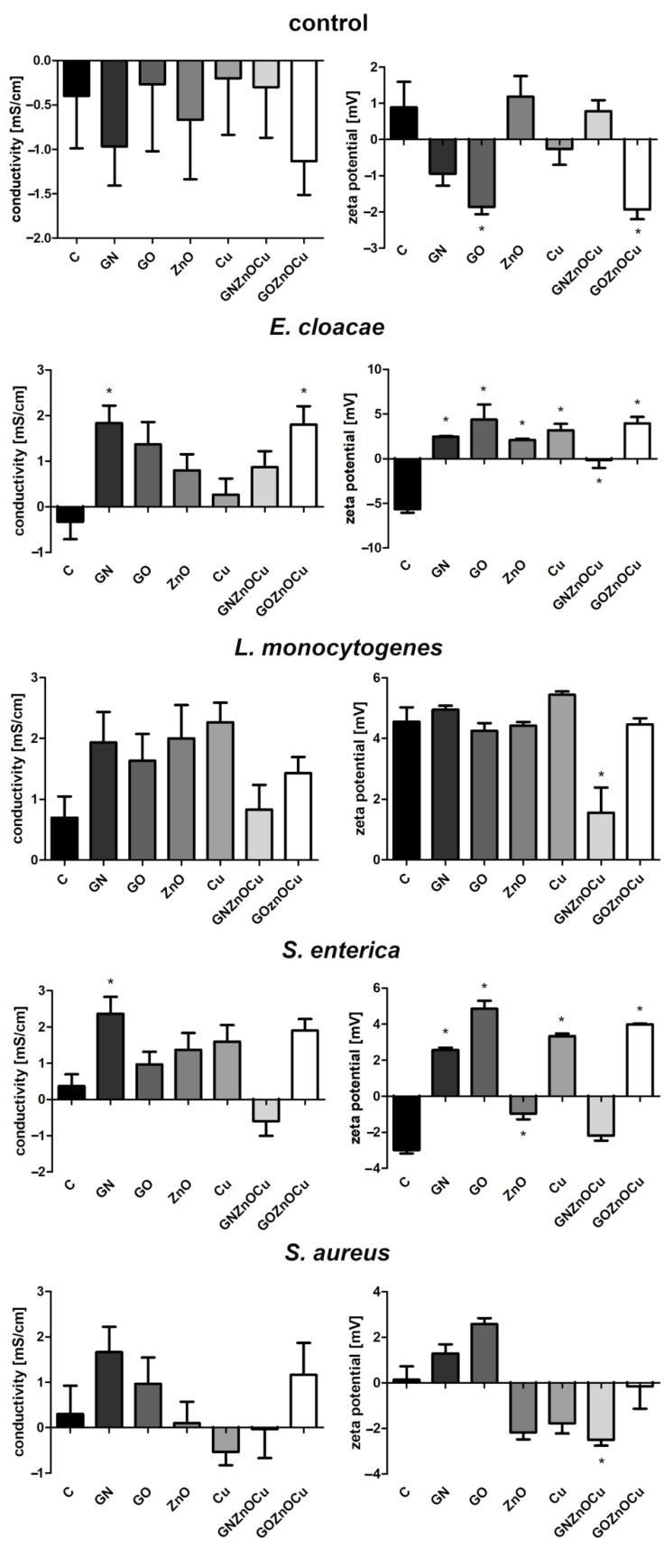
Conductivity [mS/cm] and zeta potential [mV] of bacteria species after treatment with nanomaterials. Control values are nanosuspensions without bacteria cells. C signifies the control samples, and the results are mean values ± standard deviation. “*” symbols indicate statistically significant differences in comparison to the control. The results are the difference between 0 h and 24 h of incubation with appropriate nanomaterials.

**Figure 7 nanomaterials-12-03058-f007:**
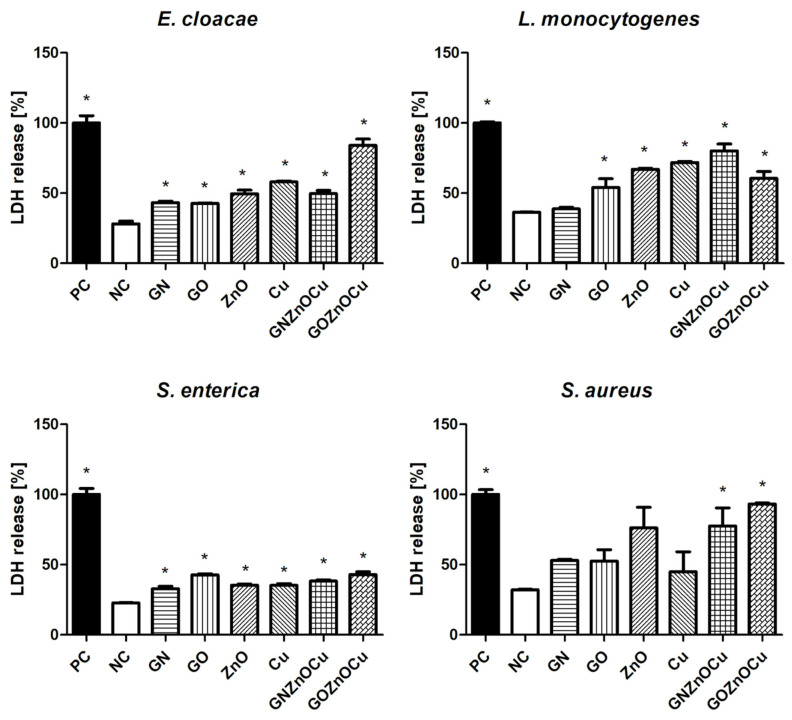
Bacteria LDH release after treatment with nanomaterials. PC signifies the positive control samples, while NC signifies the negative control samples, and the results are mean values ± standard deviation. “*” symbols indicate statistically significant differences in comparison to the control.

**Figure 8 nanomaterials-12-03058-f008:**
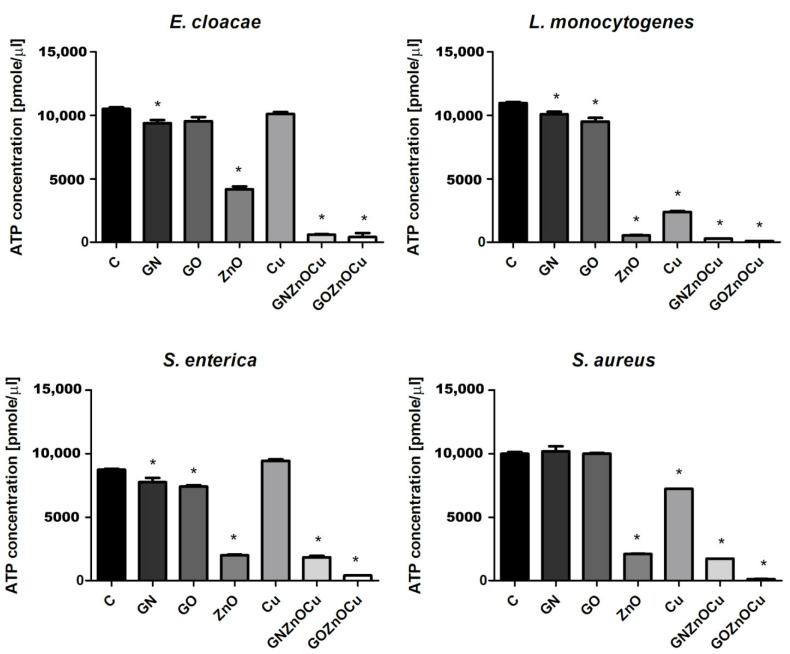
ATP content in bacteria species after treatment with nanomaterials. C signifies the control samples, and the results are mean values ± standard deviation. “*” symbols indicate statistically significant differences in comparison to the control.

**Figure 9 nanomaterials-12-03058-f009:**
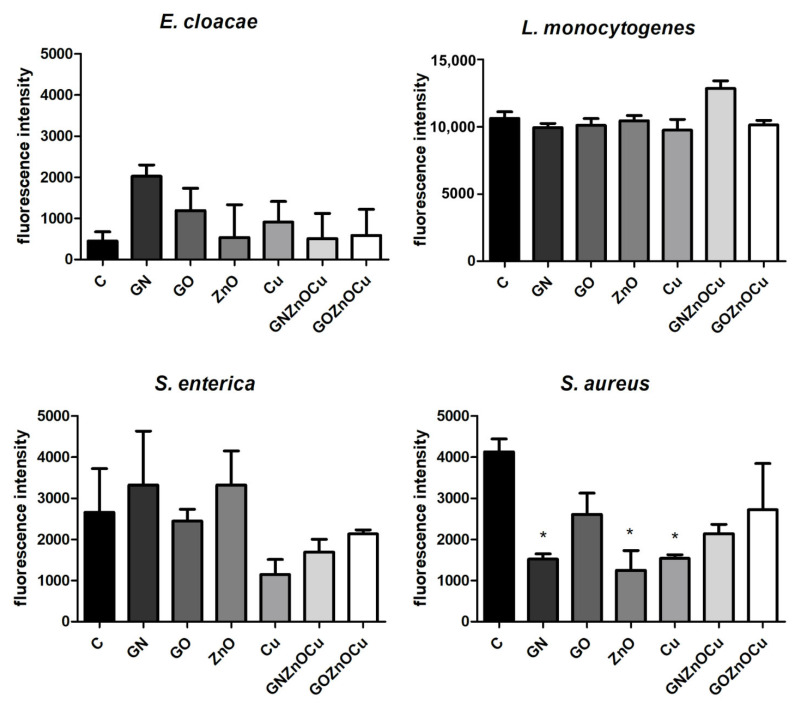
Intracellular pH in bacteria species after treatment with nanomaterials. C signifies the control samples, and the results are mean values ± standard deviation. “*” symbols indicate statistically significant differences in comparison to the control.

**Figure 10 nanomaterials-12-03058-f010:**
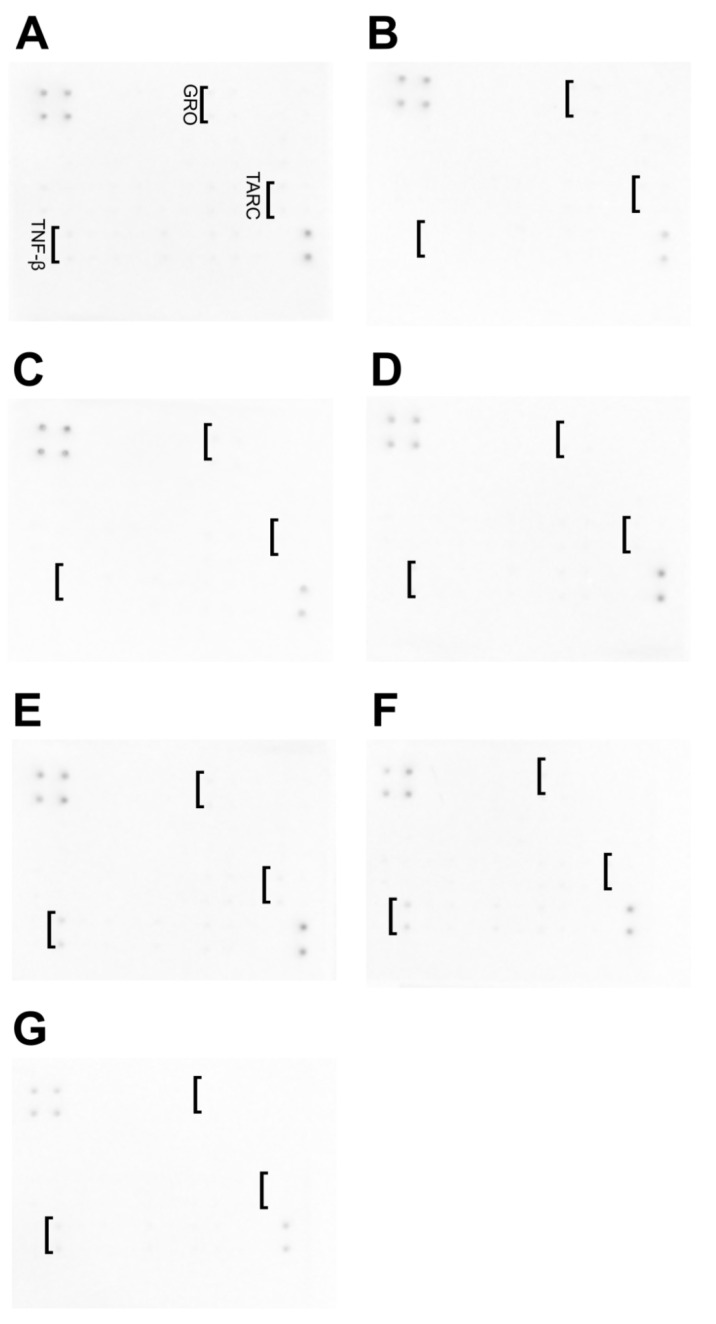
Analysis of the expression of cytokine proteins of the HFFF-2 cells after being treated with the following: (**A**) control; (**B**) GN; (**C**) GO; (**D**) ZnO; (**E**) Cu; (**F**) GNZnOCu; (**G**) GOZnOCu.

**Figure 11 nanomaterials-12-03058-f011:**
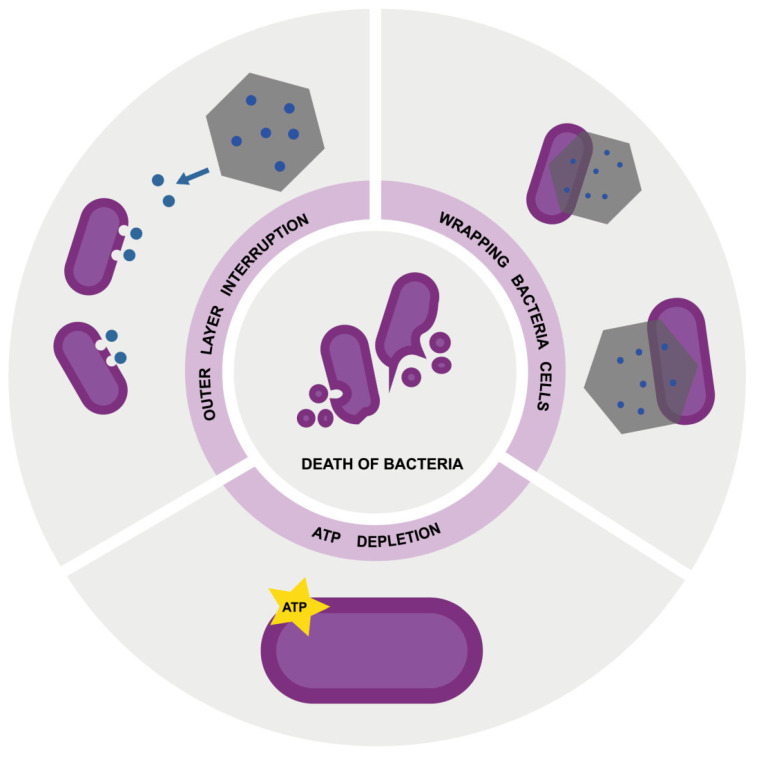
Mechanism of action of nanocomposites, which act on outer layer interruption, wrapping bacteria cells, and ATP depletion, causing cell death.

**Table 1 nanomaterials-12-03058-t001:** Mean diameter (nm) and zeta potential (mV) of nanoparticles aggregates and nanocomposites tested.

Nanomaterials	z-Average (nm)	Zeta Potential (mV)
GN	2194.33 ± 691.22	−27.40 ± 2.67
GO	614.90 ± 23.39	−27.53 ± 0.70
ZnO	2155.33 ± 148.74	2.32 ± 0.83
Cu	681.87 ± 47.18	−19.03 ± 0.51
GNZnOCu	2927.00 ± 521.67	10.56 ± 1.34
GOZnOCu	3843.33 ± 472.84	−20.47 ± 1.42

## Data Availability

The data presented in this study are available on reasonable request from the corresponding author.

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
