# Peer review of "Bacterial Surface Disturbances Affecting Cell Function during Exposure to Three-Compound Nanocomposites Based on Graphene Materials"

_nanomaterials, 2022, doi:10.3390/nano12173058_

Round 1

Reviewer 1 Report (Previous Reviewer 2)

The authors have revised the manuscript according to the reviewers’ comment. The authors have improved the resolution of the figures, including Figure 1 and Figure 3. Therefore, I would like to recommend this work to publish in Nanomateirals.

Author Response

Dear Reviewer, Thank you for your opinion.   Kind regards, SÅ‚awomir Jaworski 

Reviewer 2 Report (New Reviewer)

The authors aimed to evaluate the effect of three-compound nanocomposites (graphene or graphene oxide, Cu, ZnO) that can disrupt bacterial cells by affecting proton and electron flows. They carried out the characterization of the nanomaterials by zeta-potential measurements, hydrodynamic diameter determination, transmission electron microscope visualization and FT-IR spectra measurement. They determined the impact of nanomaterials on bacteria cells' viability and colonization, their cell wall and membrane integrity, bacterial conductivity and zeta potential, ATP concentration, and intracellular pH. The subject of the paper is interesting and worth studying, but the paper is unclear, and it is very hard to grasp the main message. The construction of sentences should be improved. Some of them are hard to understand and some do not carry any information. I strongly recommend the thorough editing of the text. There are some disputable points that need to be addressed.

The authors used the microelectrophoretic method to determine the zeta potential of nanoparticles, nanocomposites and bacteria. The claim, referring to the reference [4], that it is a method of electrostatic charge determination. It is not true. First of all, the zeta potential of any system depends on the electrolyte concentration (ionic strength) and pH of the buffer used for the suspension. In particular, the zeta potentials of oxides or proteins can strongly depend on pH. Then zeta potential is a complicated function of the surface charge and buffer composition, so one cannot claim one-to-one dependence. The authors cannot claim that they measured the charge of the cells.

Ad. 2.2 Zeta measurements of Cu and ZnO – what pH and ionic strength were used? There is no information in all text about the physicochemical parameters concerning measurements of size and zeta potential of particles or bacteria. As mentioned above, the pH and ionic strength strongly influence the value of zeta potential, and one cannot compare the values obtained in different conditions.

Ad. 2.3, 2.5 and 2.6. What was the specific reason for selecting the concentration of the nanoparticle suspensions used to treat the bacteria cultures?

Ad. 3.1. No conclusion can be drawn about the agglomeration of nanoparticles from the single value of the zeta potential. Although low absolute values of zeta potential may indicate low electrostatic stabilization (at a given ionic strength), commercial nanoparticles could be sterically stabilized. What pH and ionic strength were used to measure the zeta potential of nanoparticles?

The FT-IR spectra of GN and GO are puzzlingly similar.

In Fig.1. the size distribution by intensity is presented, but for Cu nanoparticles the size distribution by volume is given.

It cannot be said that the sizes of Zn and Cu particles are of the order of 2000 nm and 680 nm, if it is clearly seen from the TEM photos that they are aggregates, not individual particles (Fig.2). Moreover, the size given by the producer is 25 nm for Cu, and 11 nm for ZnO. The size presented in table 1 is just the size of aggregates, so do not refer to these values as the size of nanoparticles.

Ad.3.2. Not in all bacteria species, the nanocomposites were the most toxic compounds because in the case of E.cloace and L.monocytogenes the Cu nanoparticles were more toxic than GOZnOCu and GNZnOCu respectively.

Ad.3.3. The same as above – for L.monocytogenes  GOZnOCu is not the most limiting factor.

AD.3.4. The conductivity of a suspension is governed mostly by the conductivity of the buffer unless the concentration is high. The authors should separately determine the conductivities of bacteria essays and nanoparticle suspensions. What were the conditions of measurements? What was the meaning of "negative" conductivity (physically impossible)?

Ad.3.5 The authors again want to prove that the biggest amount of LDH release was observed in nanocomposite probes in all the bacteria tested without differentiating the action of GNZnOCu and GOZnOCu.  In fig.7. again the Cu nanoparticles are more reactive than GNZnOCu for E.cloace, Cu more reactive than GOZnOCu for L.monocytogenes and GO more reactive than GNZnOCu for S.enterica.

Ad. 3.7 "Intracellular pH may measure the decreases in cells treated with various substances" - decreases in what?

Ad.4. "Cu and ZnO, had diameters 682 and 2,155 nm, respectively, but both types were unstable, which was visible in the zeta potential measurements (Cu = −19.03±0.51 mV; ZnO = 2.32±0.83 mV). None of the values obtained exceeded ±30 mV, and thus, the nanoparticles are strongly cationic or anionic" – as mentioned before, a single measurement cannot indicate stability. In what sense do they have a strongly cationic or anionic character? pH dependence? Zeta potential value is rather low (conditions?), but that is not evidence of the anionic or cationic groups' strength.

"Interestingly, these types of nanomaterials have the most negative zeta values in physicochemical analysis, while in the analysis of bacterial surface charge, the samples treated with them had values greater than 0." – zeta potential is not the same as the surface charge!

"One study suggested that the GO metal substrate changed the electron transfer by absorbing electrons from bacteria respiratory pathways and modifying oxygen-containing functional groups on GO surfaces" – GO is not a metal.

"This theory confirms the pH assay, in which the decrease in fluorescence intensity was the lowest, but the pH is also influenced by the movement of efflux pumps,  which depends on ATP" – how does the intracellular pH change when the intensity of fluorescence decreases? Is it (the pH) an increase or a decrease? It will be good to explain this issue in the text.

In conclusion, I recommend the rejection of the paper in its current form and resubmission after major improvements.

Author Response

Dear Reviewer, Thank you for all suggestions. Please find the attachment,      Kind regards, SÅ‚awomir Jaworski 

Reviewer 3 Report (New Reviewer)

The authors present quite comprehensive and interesting study of nanoparticle antibacterial effect for planktonic form of bacteria, which is usually not as harmfull, as their biofilms. In this sense the study looks a bit outdated, while colloidal nanoparticles couldn't produce really fatal effect (see the modest - 2-4 orders - reduction of cfu/ml in the work) because of low-dose influence, comparing to high-dose nanoparticle treatment, e.g.,  via solid-film laser transfer [Nastulyavichus et al. "In vitro destruction of pathogenic bacterial biofilms by bactericidal metallic nanoparticles via laser-induced forward transfer." Nanomaterials 10.11 (2020): 2259]. This aspect should be discussed in the manuscript both in Introduction and Discussion section, to formulate correct novelty and specify the importance/impact of the obtained results. 

Moreover, minor changes - improvement of figures 1,3 in terms of high visiblity (like Figs.4-6) are expected.

Author Response

Dear Reviewer, Thank you for all suggestions. Please find the attachment,      Kind regards, SÅ‚awomir Jaworski 

This manuscript is a resubmission of an earlier submission. The following is a list of the peer review reports and author responses from that submission.

Round 1

Reviewer 1 Report

The article "Bacterial surface disturbances affecting cell function during exposure to three-compound nanocomposites based on graphene materials" by Lange et al. describes an extensive study of properties and ifluence of graphene/grapene oxide/znc oxide/copper nanoparticles on bacterial strains. 

The article is mostly well written, however requires some correction of style and language especially in the abstract and introduction parts. The experimental physicochemical studies are well performed and presented. Unfortunately, the whole article is not suitable for publication, since the claimed antimicrobial activity of the described materials is apparently non-existing. The viability numbers obtained by authors and given on figure 4 are within 15-80%, which is way too high.  For a proper antimicrobial action, the decrease in CFU after the treatment must be at least 3 log, and even though viability is not always directly comparable with the CFU counting, the viabilities obtained for the nanocomposites are still too high. Having said that, the rest of the article simply makes no sense.

Reviewer 2 Report

The authors have demonstrated antibacterial mechanisms for three graphene-based nanocomposites. To investigate structural and optical properties of three graphene-based nanocomposites, hydrodynamic diameter, zeta potential, TEM image, and FT-IR analysis were performed. The authors have also demonstrated the antibacterial activities of three graphene-based nanocomposites based on viability, conductivity and surface charge, cell wall integrity, ATP concentration, and intracellular pH. Overall, this work can inspire more material design ideas of the graphene-based nanocomposites for antibacterial application. Therefore, I would like to recommend this work to publish in Nanomateirals. Below are some comments for the authors.

1. The abstract needs to be revised. For example, the sentence “The aim of the study was to evaluate the mechanism of three-compound nanocomposites based on graphene materials”. What mechanism did the authors study? For the sentence “To determine the nanomaterials’ properties, an analysis of mean hydrodynamic diameter and zeta potential, transmission electron microscope (TEM) visualization, and FT-IR analysis were performed.”, What the properties did the authors determine? Furthermore, there is a strange symptom “ ‘ “ in the sentence.

2. Figure 1 is too blurred. The authors should provide higher resolution images.

3. In Figure 3, FTIR spectra are also too blurred. The authors should provide higher resolution spectra.

4. The characteristic peaks of samples should be labeled in the FTIR spectra.

5. For the introduction “Nanomaterials are used as antimicrobial agents because of multiple mechanisms that...”, more references could be cited to broaden the introduction.

https://doi.org/10.3390/ijms20122924